# Decision-making approaches used by UK and international health funding organisations for allocating research funds: A survey of current practice

Katie Meadmore[1]*, Kathryn Fackrell[1], Alejandra Recio-Saucedo[1], Abby Bull[1], Simon D. S. Fraser[1,2], Amanda Blatch-Jones[1]

1 Wessex Institute, University of Southampton, Southampton, United Kingdom, 2 School of Primary Care, Population Sciences and Medical Education, Faculty of Medicine, University of Southampton, Southampton, United Kingdom

* K.Meadmore@soton.ac.uk

**Data Availability Statement:** All relevant quantitative data are within the manuscript and its Supporting Information files.

## Abstract

Innovations in decision-making practice for allocation of funds in health research are emerging; however, it is not clear to what extent these are used. This study aims to better understand current decision-making practices for the allocation of research funding from the perspective of UK and international health funders. An online survey (active March-April 2019) was distributed by email to UK and international health and health-related funding organisations (e.g., biomedical and social), and was publicised on social media. The survey collected information about decision-making approaches for research funding allocation, and covered assessment criteria, current and past practices, and considerations for improvements or future practice. A mixed methods analysis provided descriptive statistics (frequencies and percentages of responses) and an inductive thematic framework of key experiences. Thirty-one responses were analysed, representing government-funded organisations and charities in the health sector from the UK, Europe and Australia. Four themes were extracted and provided a narrative framework. 1. The most reported decision-making approaches were external peer review, triage, and face-to-face committee meetings; 2. Key values underpinned decision-making processes. These included transparency and gaining perspectives from reviewers with different expertise (e.g., scientific, patient and public); 3. Cross-cutting challenges of the decision-making processes faced by funders included bias, burden and external limitations; 4. Evidence of variations and innovations from the most reported decision-making approaches, including proportionate peer review, number of decision-points, virtual committee meetings and sandpits (interactive workshop). Broadly similar decision-making processes were used by all funders in this survey. Findings indicated a preference for funders to adapt current decision-making processes rather than using more innovative approaches: however, there is a need for more flexibility in decision-making and support to applicants. Funders indicated the need for information and empirical evidence on innovations which would help to inform decision-making in research fund allocation.

**Funding:** This study is supported by the National Institute for Health Research (NIHR) Evaluation, Trials and Studies Coordinating Centre (NETSCC) through its Research on Research programme in the form of salaries for all authors. The views and opinions expressed are those of the authors and do not necessarily reflect those of the NIHR or Department of Health and Social Care.

**Competing interests:** All of the authors are employed by the Wessex Institute, University of Southampton and work within the National Institute for Health Research, Evaluation, Trials and Studies Coordinating Centre (NETSCC). SF is also employed by the University of Southampton as an Associate Professor of Public Health and KF also holds a Post-Doctoral Fellowship funded by the NIHR. This does not alter our adherence to PLOS ONE policies on sharing data and materials.

## Introduction

Health research funding organisations have to make difficult decisions regarding which research applications to fund. For example, deciding which health areas or research questions have priority, and whether the research may lead to changes in practice and better health outcomes (be that patient, economic or social benefit). In theory, funding decisions should try to be objective, assessing all applications against criteria in a fair, consistent and transparent way, and those applications deemed to meet the assessment criteria should be funded. However, in reality, funding decision-making is much more complex and often requires balancing evidence-needs, assessment criteria weightings, potential impact, workload capacity of funding organisation staff, applicants and reviewers, and a finite financial resource [1]. The challenges to funding decision-making are well established and numerous, and although innovations in decision-making are emerging, it is not clear to what extent these are utilised by funders.

The process of decision-making is often facilitated by peer review [2–4]. Peer review refers to a process by which an application is assessed by an expert in the research area, a person with related expertise (e.g., academic, clinician, health economist, methodologist, patient) or a member of the public. From the perspective of research funders, this can include external peer review, in which the reviewer is independent from the funding organisation and provides a written review, and/or a funding committee that reviews and discusses the application and is often built into the funding organisational process. Peer review, and specifically external peer review, is often considered critical for the decision-making process [3,5], with a recent report finding 78% of survey respondents agreeing that peer review was the best method by which to allocate research funds [4].

There are many benefits to using peer review including receiving expert opinion on particular ideas, methodologies, practicalities and potential future implementation of the research. There are also benefits to the reviewers themselves including keeping up to date with progress in their fields which helps to enhance personal development and can feed into teaching and research practices [5]. However, it is also well established that peer review has flaws including being susceptible to bias. For example, it has been shown that peer reviewers and funding decisions have favoured applicants of specific ages, career stages, genders and institutions, amongst others [6–11]. Furthermore, peer review is inherently subjective. Peer reviewer scores and recommendations often vary widely [1,3,7,11–15] and do not always predict that the research will be successful or will have impact [8,16,17]. Peer review is also considered burdensome, and there is a high cost in terms of time and financial expense for applicants, reviewers and funding organisations [3,8,18–21].

Despite the known challenges of peer review, the same processes and associated issues are still largely reported [22]. Variations and innovations in decision-making are emerging to tackle the issues of bias, burden and cost [5,11,23,24]. For example, variations to application forms, numbers of reviewers, teleconferencing and innovative approaches such as sandpits (an interactive workshop over a number of days whereby stakeholders with an interest in research on a particular topic are encouraged to collaborate on innovative solutions to a research question) and modified funding lotteries [11,13,25–29]. However, empirical research activity in these areas is limited and challenges in conducting rigorous testing of innovative approaches has resulted in a limited evidence base for decision-making approaches [30].In addition, it is not clear to what extent these are used by health funding organisations. For example, in 2011, the European Science Foundation conducted a survey [31] to explore peer review practices, with the focus on quality assurance of reviews, identifying, incentivising and managing data on reviewers, and how proposals are managed. The survey did not report detailed information on the types of processes used in peer review practices nor whether any innovative approaches

were being considered by health research funders. Identifying what approaches are used in current practice in decision-making and whether they include innovative approaches, may provide better understanding about decision-making and why funders are engaged (or not) in exploring mechanisms to change/enhance funding processes to address known challenges.

The UK National Institute for Health Research (NIHR) Research on Research (RoR) team are addressing the lack of an empirical evidence-base in a programme of work exploring peer review and its role in the decision-making process for the allocation of research funding. This paper reports the results of the first study to be completed in this programme of work. The aim of the study was to identify and explore decision-making practices used by UK and international funders to better understand the current decision-making landscape for the allocation of health research funding.

## Materials and method

### Design

This study used a survey design to gather quantitative and qualitative information about decision-making practices used by organisations that fund health and health-related research. The survey used closed, tick-box questions and open (free-text) questions and was designed to be delivered online in order to have national and global reach. Open questions were underpinned by a qualitative phenomenological approach that aimed to explore and build understanding of experiences in decision-making processes from the perspective of the funding organisations. The study was approved by the University of Southampton, Faculty of Medicine Ethics Committee (ERGO ID 46851, February 2019).

### Survey development

The survey was delivered online using iSurvey software maintained by the University of Southampton (https://isurvey.soton.ac.uk/). The questions were developed using an iterative process involving NIHR staff and members of the NIHR, Research on Research team. Initially, the authors compiled a list of potential questions for inclusion in the survey, which were grouped into sections. Questions were developed based on discussions with the team and from previous NIHR projects [23,28,32] and existing literature about different types of decision-making processes (e.g. peer review, triage, sandpits; [31,33]) and were distilled down to generate a 27-question survey covering three sections.

The survey was piloted with four members of NIHR staff known to the research team to determine the relevance of the proposed questions, the face validity of questions, in particular language, comprehension and completion time, and the construct validity of the questions and response options. Two members of staff provided written feedback and two members of staff provided verbal feedback as they tested the survey. Feedback was used to refine and re-order some of the original questions and consensus was reached by agreement from the research team.

The final survey consisted of 27 main questions (see S1 File). Respondents were asked to identify one research programme or funding call within their organisation that they would focus on to complete the survey. They were asked to choose a funding programme or theme (for research projects or programmes, not fellowships or infrastructure) that they were most familiar with. It was made clear that respondents could complete the survey multiple times (for different funding programmes). Each section is described in Table 1.

**Table 1. Description of contents of each of the three survey sections.**

| Section | Number of questions | Description of questions |
|---|---|---|
| 1 | 13 | Characteristics of the funding organisation |
| 2 | 10 | Current decision-making practices and if/how these could be improved; Practices used in the past; Benefits and drawbacks to these systems |
| 3 | 4 | Decision-making practices that funders might be interested in exploring in the future and why |

## Distribution of survey

Purposive and snowball sampling was used to recruit respondents. In order to obtain a broad funder perspective 76 health research funding organisations (109 targeted emails) were contacted across 10 different countries. Where possible emails were sent to named administrative staff for the research programmes, but also included general enquiry email addresses and online forms. Other than some colleagues at the NIHR, there were no prior relationships with potential respondents.

The targeted list of organisations was collated through an online search for health funders and included charities, research councils and other government funded organisations in the UK and internationally. The online search was complemented by cross-referencing with collated lists of funding organisations on websites and in reports (e.g., [34,35]), as well as through known contacts of NIHR staff. Organisations were considered eligible if the remit of research that was funded was health or health-related research projects or programmes (not fellowships or awards funding an individual person or infrastructure).

The survey was launched on 6th March 2019 and was open for seven weeks, closing on 17th April 2019. The survey was also promoted using e-promotion routes including blogs on the Association of Medical Research Charities [36] and NIHR websites [37] via social media channels such as Twitter (e.g. NIHR twitter account), and through other organisational distribution lists and newsletters (e.g., The International Network of Agencies for Health Technology Assessment and Health Research Authority). The survey was also promoted at national and international conferences (e.g., Ensuring Value in Research (EViR) Funders' Collaboration and Development Forum, March 2019). Potential respondents either received an invitation letter and link to the survey via email or could access the survey link via social media posts. The survey link took respondents to an information sheet and required them to provide online consent before the survey questions were displayed. Two reminder emails and tweets were sent, one at two weeks before and a second two days before the survey was due to close.

## Data analysis

A mixed methods approach to the analysis was taken in order to cross-validate findings and provide a fuller picture of the peer review landscape. Data were analysed separately but concurrently, and findings from each strand of data were triangulated to inform and explain patterns and interpretation. For example, qualitative data was used to expand and interpret quantitative findings and frequency data was used to confirm patterns in qualitative data.

Prior to analysis, data was screened as part of data cleaning by at least two members of the team. If multiple responses were received from one research programme, the responses were merged so that there was only one entry for that research programme. All data was stored in a dedicated research folder on the University of Southampton's internal secure server.

For the quantitative data analysis, survey data was downloaded to Microsoft Excel 2016©. Descriptive statistics were used to identify frequencies and percentages of responses to closed

questions in relation to characteristics of funders, assessment criteria and types of peer review processes engaged in. Due to the small sample, it was not considered valid to conduct further analyses, such as regression techniques or comparative analyses to explore factors associated with the adoption of particular peer review methods used by funders (e.g., inter-country differences).

For the qualitative analysis, redacted PDFs of survey responses were uploaded to NVivo 12. Free text responses were subjected to inductive thematic analysis in NVivo12 using the 6-step framework [38]. In the initial stages, KM read all the survey responses from all respondents to allow for familiarisation of the responses (step 1). Where there were limited responses, a discussion was held with members of the research team to gain consensus about whether to exclude or include the response. Open text was then coded into simple words or phrases that described the topic of the sentence or word (e.g., "bias", "transparency"; step 2). These initial codes were refined and grouped together to form themes and subthemes (step 3). These initial themes were then reviewed and discussed with the team and regrouped, refined and defined through an iterative process (steps 4 and 5). The coding process was inductive as no prior framework was considered. However, the authors were mindful that the data came from pre-defined survey questions and tried to take a deeper interpretation of the data and did not just group the codes under the question headings. The COREQ Guidelines were adhered to in the reporting of the qualitative data as a quality check [39].

## Results

### Respondents

A total of 35 responses were received from respondents in 24 different funding organisations (see Fig 1). For the quantitative data, 31 responses (from 23 funding organisations) were analysed. Multiple entries for two research programmes were merged. The most complete entry was kept, and any blank responses were filled using the other entry. Open responses were combined. No conflicts in open responses were observed. Closed question conflicts were resolved through discussion and by checking the funder website. An additional two responses were excluded as the respondents indicated that they were not the appropriate person to complete the survey and so may not have provided a true reflection of the organisation processes. For the qualitative data, three additional responses were excluded from analysis as no open questions had been completed. Not all respondents provided answers for all questions. For six funding organisations, more than one response was submitted and these pertained to different research programmes. Respondents completed the survey in an average of 33 minutes (SD = 18 minutes).

The respondents represented funding organisations in a broad range of health areas, including ageing, neurodegenerative diseases, cancer, diabetes, meningitis, health technology, HIV and AIDS, heart disease and stroke. They covered basic science through to applied clinical research as well as health service delivery, and included disease-specific programmes, public health and global health.

The majority of responses (18/31, 58%) were for research programmes of funding organisations in the charity sector, with research councils and other government-funded organisations also being represented (13/31, 42%; see also S1 Dataset). The majority responses were from funding organisations based in the UK (23/31, 74%), with smaller numbers from Europe (6/31, 19%) and Australia (2/31, 6%). The size of the funding organisations ranged from 0–9 staff for charities and 25–99 staff for government funded organisations to over 250 staff. The average amount funding given per award ranged from £25,000 to £4.5 million, the average number

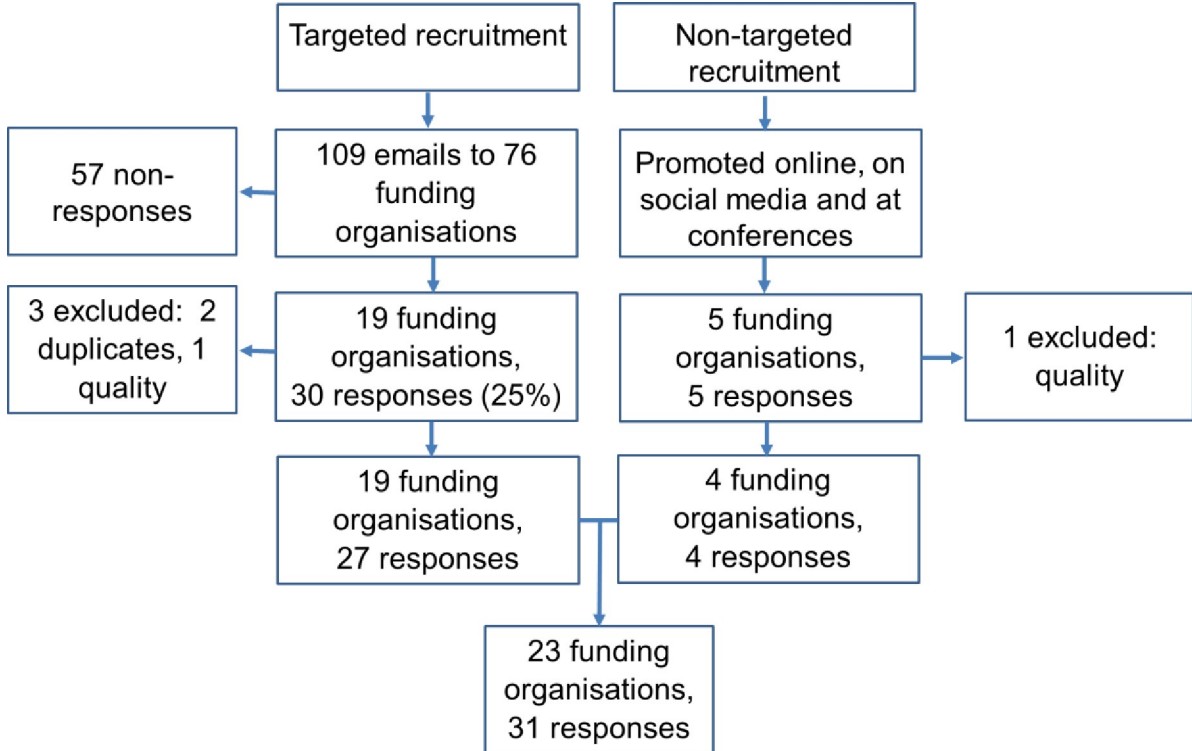

**Fig 1. Diagram to show response rates.** Quality = respondent identified as not being the correct person to complete the survey and did not provide complete responses. Funding organisations could provide more than one response (for different research programmes). Duplicates = There were multiple entries for a research programme. These were merged so that there was only one entry per research programme.

of applications received per round ranged from 10 to 2500 (median = 42 applications/round) and success rates of the funding programme ranged from 3–100% (median = 20%).

## Peer review data

Four key themes were extracted from the qualitative data, and for narrative purposes the findings from the quantitative and qualitative data are integrated together and described under these themes. Quotes are written verbatim, although some words have been redacted to maintain confidentiality. Participant number and funding source are provided for context for the quotes.

**Theme 1. "Current landscape": Typical decision-making processes.** This theme referred to the reported decision-making processes that were typically used (see S1 Dataset). The data show that nearly all research programmes used some sort of triage (24/29, 83%), external peer review (28/29, 97%) and face-to-face committee meetings (25/29, 86%; see Fig 2). However, these were not the only processes used (and in many cases they were used along with variations to these processes; see theme 4). The use of these processes in the typical review pathway were seen as essential to support decision-making through reducing volume of applications, encouraging discussion and engaging experts.

*The face to face funding committee meetings enable the opportunities for discussion where a decision is not clear cut.* P33, Government.

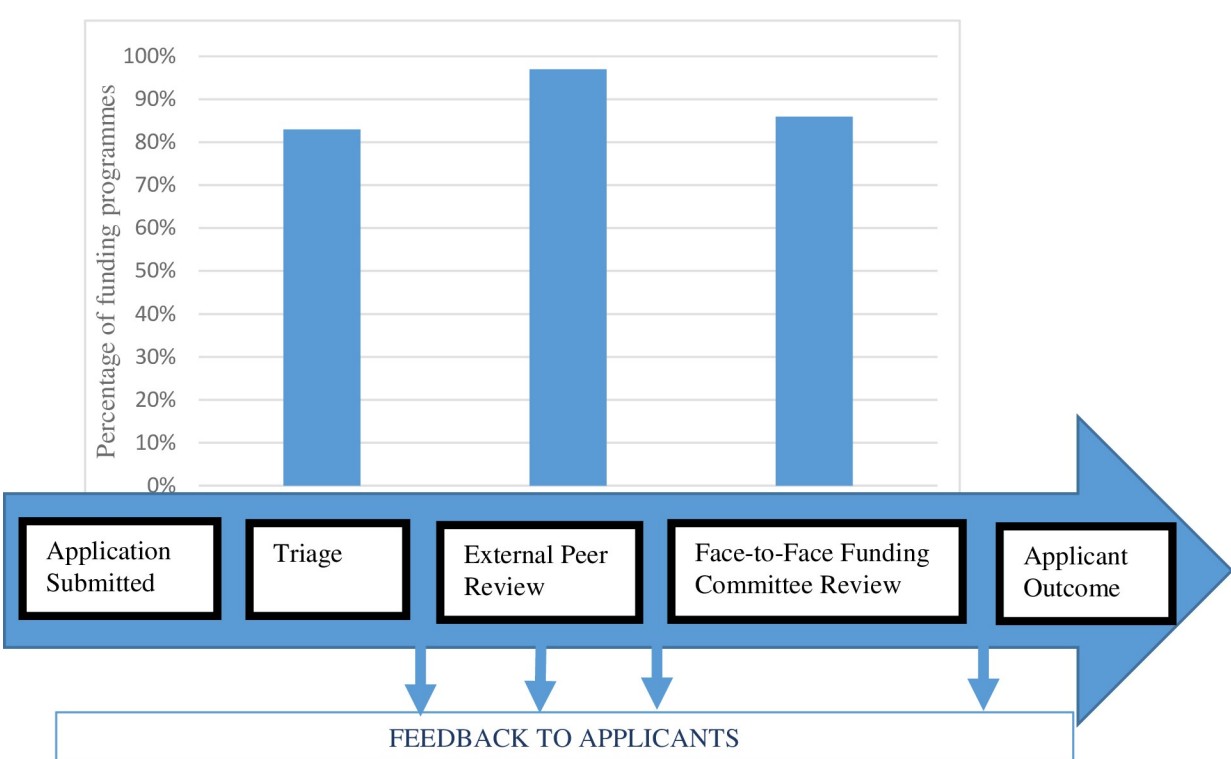

**Fig 2. A typical decision-making pathway and percentage of respondents that use the process.** Total number of survey responses = 29 (two entries were blank so not included in analysis).

*The process involves external experts in the field which should ensure the funding organisation takes into accounts the latest development.* P5, Charity.

*As we cover all areas and types of [disease-type] research, we need external peer review to ensure expert review of all applications.* P9, Charity.

Within these processes, respondents also discussed the importance of feedback to applicants and the right to reply. Respondents indicated that scores as well as written comments by peer reviewers were provided to applicants in order to help strengthen the application for future rounds or submissions. However, the challenges of providing feedback were also acknowledged.

*Applicants have feedback at several points about how to strengthen their proposal, which assists with deliverability and start up of projects.* P22, Government.

*Reasoning for not-funding is challenging to communicate and is not always recognised by applicants.* P24, Charity.

The most common scoring system that was used to aid decision-making was numeric (27/29, 93%), although four respondents qualified that the numeric scores also had written qualitative descriptions for clarity. Scores ranged from a 4 to a 20-point scale. Scores were given by external peer reviewers and/or committee members (28/29, 97%), and scoring systems were not always the same for these two groups. For example, one respondent reported that external peer reviewers use a 1–6 scale and the committee use a 1–10 scale. Organisations that used

committee meetings used mean, median, consensus and voting to make a final decision, and often used a combination of methods.

**Theme 2. Values underpinning decision-making processes.** This theme relates to the values that funders see as either integral to current practice or as important to engage in for future practice consideration. All funders, and research programmes within funders had different strategic objectives and underlying aims, and these were reflected in the assessment criteria (see Fig 3 and S1 Dataset).

> *In this program, pilot projects are funded. Therefore, there is a high risk and possible high gain connected to the funded research. Other programs not involving pilot projects have moderate or low risk and [are] more focussed on timeliness and the highest societal impact.* P29, Charity.

Additional criteria that had not been pre-listed in the survey were also reported. These tended to fall under two broad areas; namely the team (focus on track-record and career development, as well as multidisciplinary and international collaborations) and the project (ambition of the project, intellectual property and commercial strategy, and methodological development).

Respondents indicated that the decision-making processes that their organisation used were done so because they demonstrated positive values of decision-making, including transparency, fairness, robustness and rigorousness.

> *The use of a set process allows for more transparency and fairness, all applications being treated according to the same criteria.* P5, Charity.

Respondents also commented on the need to have different stakeholder perspectives (academic, clinician, patient) reviewing the applications. For some organisations this is standard practice, whereas for others it was something that the organisations wanted to engage in, in the

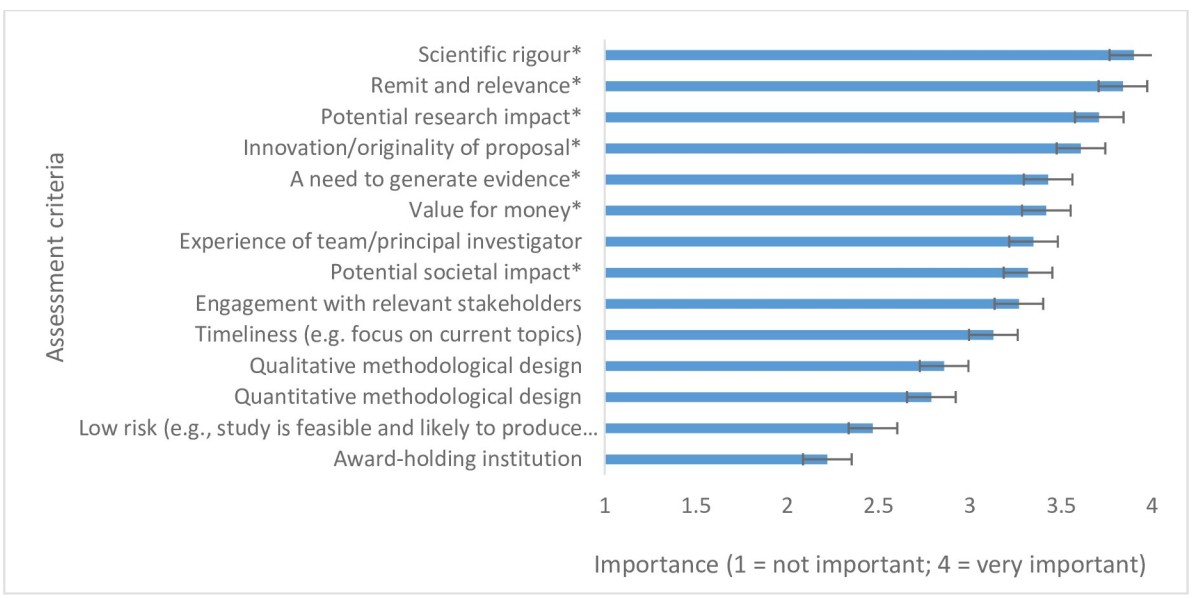

**Fig 3. Assessment criteria based on mean scores for importance.** 1 = not important and 4 = very important; * denotes those criteria that averaged a median score of 4; Total number of survey responses = 31.

future. Regardless of whether this was already integral to current practice, ensuring different perspectives was seen as an important feature of review practice.

> *External peer review and panel assessment by academics, research users and patients/public provides a range of informed perspectives.* P10, Government.

> *We also have review by people affected by [disease population] to ensure the research is relevant to our beneficiaries.* P8, Charity.

> *Public review is currently still outside the decision-making, but will be brought in over the next years.* P20, Government.

**Theme 3. Challenges of the decision-making process.** This theme relates to the cross-cutting challenges in decision-making approaches that the funder faces. The main challenges that were reported were bias and burden, with 89% of respondents (25/28) commenting on these issues. These comments encompassed both bias and burden as disadvantages of the decision-making process, as well as ways in which they could be or are being managed. Challenges were mainly drawn from the open survey questions asking about the benefits and drawbacks of decision-making approaches.

Bias was described in relation to peer review generally, as part of external peer review, committee review, and in scoring. Respondents acknowledged that processes had potential for bias towards particular people, groups or projects.

> *There is a propensity for low risk projects to be funded as the project eliciting wide opinions and scores can be scored lower when averaged.* P23, Charity.

It was also recognised that there could be inconsistency between peer reviewers' feedback.

> *Peer reviewers can sometimes vary in opinion about the same application—that's why we prefer 3 reviews per application.* P2, Charity.

> *Scoring, even when supported by criteria, can be very personal. Some scorers are more generous than others (a 3 may mean a 4.5 from somebody else and vice versa).* P5, Charity.

When this happened, comments suggested that this could result in reviews being ignored, *"too much variability in quality of reviews meaning they were generally ignored"* (P11, Charity), or that *"Opinions [of other reviewers] can be swayed by strong opinions"* (P23, Charity). Solutions to reduce this bias included stopping certain processes, engaging in better training or education for reviewers, recruiting strong Chairs, undertaking double-blind or open peer review, and using both scores and written comments as "*the scores do not always reflect the comments*" (P15, Government).

Burden was discussed in relation to monetary cost, time and workload for all stakeholders.

> *It is a lot of work for the applicants, the external peer reviewers, the secretariat staff, and the committee members.* P22, Government.

In terms of workload, some respondents commented on the difficulties in obtaining and/or securing potential peer reviewers, especially when trying to find people with the relevant expertise. Challenges with finding peer reviewers is further complicated by trying to obtain the appropriate number of people to review applications often resulting in managing conflicts of

interest. Face-to-face committee meetings were also seen as resource intensive, with additional cost burden to the funder. The fact that face-to-face meetings took a number of days, had high paperwork loads, and included multiple committee members meant that they were difficult to arrange and prepare for. These time burdens were undesirable, prolonging final decisions and detracting from the research programme.

*Face to face meetings with multiple experts are hard to arrange. Finding a panel with the right expertise and no conflicts of interest is extremely difficult*. P26, Charity.

*Peer review process takes too long and the associated timelines make the funding programme less attractive to small-to-medium enterprises*. P28, Government.

There was also discussion surrounding the number of stages and types of application that were used in relation to burden. Some regarded one stage processes as more efficient in terms of time allocation compared to a two stage process. The two stage process was more debatable, with some respondents indicating that the two-stage process increases burden in relation to time and effort to all stakeholders, whilst other respondents felt that a two-stage process may reduce burden.

*Only applicants who pass shortlisting stage have to do all the work involved in a stage 2 application*. P22, Government.

*Seeks to optimise time taken for teams to develop applications and in reviewer time by using two stages*. P19, Government.

These differences in opinion are an indication of why there is variability in the number of stages and decision-making points used by funders. This variability was further complicated by external limitations and the regulations that organisations have to comply with.

*As part of [an organisation] we are obligated to ensure we follow certain principles and practices and we are audited accordingly*. P3, Charity.

In addition, one participant reflected on the fact that funders are often reliant on voluntary contributions from external peer reviewers and committee members.

*We are very reliant on other people's time—to provide reviews and to participate on the panel. When they delay sending the promised review, or if people cancel attendance at the Panel, you can't moan as they are all volunteers*. P2, Charity.

**Theme 4. "Emerging Landscape": Not one size fits all.** The final theme describes the emerging landscape in decision-making processes and demonstrates that there is '*not a one size fits all*' for the allocation of funding decision-making processes. The key concept was the need for funders to be flexible in their funding approaches and to be more supportive of applicants and reviewers. Funders acknowledged that not all approaches work in all research areas, and that there is a need to foster more collaboration and flexible thinking to uphold core values, fund the right research and maximise reviewer contributions.

*We are always open to different ways of working and recognise that there is not one-size fits all approach to decision-making. For example, sandpits work for some research areas but not all*. P15, Government.

By funders taking a more flexible and supportive role, respondents indicated that this would help to ensure that the applications were tailored to the funders strategic aims and requirements and may enhance opportunities for success.

*Potential for new ways to iterate with teams where there is a good question but significant issues to resolve*. P19, Government.

*As an organisation, we want to encourage greater collaboration—this could be relevant to us in the future when we plan to commission research rather than use a response-led approach.* P7, Charity.

It also highlighted that more support could be offered to reviewers both in relation to training and incentives, and that by doing this it may make reviewing grant applications a more attractive prospect.

Comments reflected that many funders already engage in continuous improvements for decision-making, whilst others were aware of work that has been done by others. It is important to note, however, that not all respondents (19/28, 68%) agreed that their current decision-making approach needed to be improved.

In support of '*not one size fits all*', the data showed that funders are using variations to typical processes (see Fig 4). Variations to typical processes encompassed variations to external peer review, committee meetings, applicant input or decision-stages within the decision-making pathway. However, respondents also commented that they would like to know more about variations and innovative approaches. As described in theme one, those who used variations to typical processes, usually did this in addition to one or more of the typical processes. For those who did not use face-to-face committee meetings, they either used virtual committee meetings or a sandpit approach (see Fig 4). However, these approaches also have pros and cons.

*No external peer review, only assessment by panel within the Workshop, so novelty (overlap with existing work) may be harder to judge within short-frame of the idea generation and 'pitches'.* P25, Charity.

## Discussion

The aim of this study was to survey UK and international funding organisations to better understand current decision-making practices employed for the allocation of health research funding. In line with previous work [3,8], the typical pathway for allocation of funding remains inclusive of triage, external peer review and face-to-face committee meetings. However, there are many variations to this typical pathway, with nuances for different research programmes and/or funders (e.g. different numbers of decision-points, proportionate review). It was clear that funders engage in these processes because they believe them to demonstrate positive values that are important to stakeholders, such as transparency, quality, patient benefit, as well as providing a framework to objectively assess applications according to criteria that match the funder's strategic goals. Indeed, respondents were keen to emphasise these values as being the main benefits of the decision-making processes used. This may highlight the pressure on funders, particularly public funders, to be accountable and demonstrate the added value and benefit to society for the funding decisions that they make. However, despite our findings indicating that funders believe they have transparent processes and the importance of transparency, a recent report that surveyed grant applicants showed that they were least satisfied with this aspect in the funding pathway [4].

| • **Variations to typical process** |
|---|
| **External peer review variation** |
| Proportionate external/written peer review (where the number of reviewers are assigned according to the size of the grant or where reviewers are assigned certain sections of the application, based on expertise) |
| Open peer review (reviewers comment publicly on applications) |
| **Committee variation** |
| Flexible board/committee (different members attend each meeting based on expertise) |
| Virtual board/committee meetings (members 'meet' via telephone or an online platform) |
| **Applicant variation** |
| Rebuttal to feedback (applicants get a chance to revise their application and reply to peer review comments) |
| Interviews and presentations |
| **Process variation** |
| Number of decision-stages |
| Partial random allocation/lottery (applications to be funded are selected at random from a predefined pool)* |
| • **Alternative processes** |
| Sandpits (groups of independent academics/clinicians/researchers are invited to collaborate on research projects) |

Typical processes only (21%)

Variations to typical processes (69%)

Variations and alternative processes (3%)

Alternative processes only (3%)

**Fig 4. Types of different processes used.** Variations to typical processes and approaches that had some uptake from funding organisations and Venn diagram to show percentage of research programmes who use typical, variations to typical and innovative approaches (from total survey sample of = 29). * This process had interest but not uptake (within the survey sample).

In line with previous work, the findings showed that bias and burden are considered to be the biggest challenges of the typical decision-making pathway (e.g., [3,4,8,10,11,19]). However, the survey highlighted that funders are aware of these challenges and some are taking additional steps to try and overcome them. Methods for doing so include increasing diversity of reviewers by including patients and public members as external peer reviewers and on committee panels, and using teleconferencing rather than face-to-face meetings to increase the potential reviewer pool and to cut down on meeting time and costs. However, these solutions also come with their own challenges. For example, health research covers a broad range of topics and so what is considered expertise will also differ across applications and funders. In practice, funding organisations cannot feasibly cover all areas for each application (especially when trying to reduce burden) and therefore need to balance number of reviewers with which areas of expertise are priority.

The survey also highlighted that funding organisations have very limited control on many aspects of the peer review system, and are reliant on the contributions of reviewers and committee members. Such limitations can impact the amount of time required for a decision to be made, and it is perhaps important for all stakeholders to keep this in mind when considering the decision-making pathway.

Whilst it was clear that there is 'not a one size fits all' for peer review practice across different funding organisations and research programmes, the nuances also demonstrate differing opinions on what is considered best practice. For example, there was debate on whether one or

two stage processes increase or reduce burden, and there was large variability in scoring scales. These differences also reflect the variability in assessment criteria and strategic goals across the funding organisations. In addition, there are practices that funders acknowledge as being valuable, such as providing written feedback to applicants and allowing rebuttal, but the issue of burden and limited resource constrains some funders from doing this. Indeed, Langfeldt [40] found that funding decisions by committees were most influenced by budget restrictions and the type of scoring method that was used. It is the variability in these processes that have most influence on funding decisions [40].

There was acknowledgment that funders need to be more flexible in their approaches to decision-making, providing more support to applicants and encouraging greater collaboration between applicants, and applicants and funders. Previous work has suggested that funders provide poor support for applicants [41], and burden associated with grant funding falls heavily on applicants as well as reviewers and funding organisation staff [3,42,43]. Although it is appropriate to challenge applicants to ensure rigorous and reliable research, by taking a more flexible and supportive role, and through fostering collaboration and working more closely with applicants, quality, innovation or other important values to funders may be enhanced. This may increase stakeholder (e.g., funder, applicant) burden in the short-term but over the longer term may result in better quality applications, project outcomes and impact. This has also recently been recommended by the AMRC [44]. This is an area for consideration and evaluation and evidence is needed to determine whether this approach will work for funders and researchers and where additional burden may lie.

Many funders seem to be actively engaged in continuous improvement of their decision-making practices. However, the many variations to the typical pathway indicate a preference for funders to make small adaptations to current systems rather than employ innovative approaches such as sandpits and lottery systems. Research has shown conflicting results over different approaches to decision-making [30] which may make funders wary of trying new processes. In addition, there are still very few good quality empirical studies that have evaluated specific decision-making approaches and funders need this evidence base to inform their practice. Moreover, the findings from this survey indicated not only a need for evidence but also a need for more information about innovative approaches. Funders may be more interested in innovative practices once more information (what the process is and how to implement it) and evidence is available. This may also vary across different funding organisations as implementing a new peer review approach would potentially cause significant strain to the funder who would also be trying to maintain a high standard of practice.

This survey has several limitations. The results reported are based on a relatively small number of responses (N = 31), which are open to bias as responders were self-selected, and so caution must be applied in interpretation and generalisation [45]. However, 30 responses was our target sample size and represented a range of different organisations and countries. Note also that five respondents were recruited via the use of e-promotion and promotion at conferences, demonstrating the benefit of multi-modal recruitment routes [46]. Multi-modal recruitment methods are recommended to researchers looking to recruit international or multidisciplinary samples [46]. We also sent reminders and stated the average time it would take to complete the survey, and these strategies are suggested to improve online survey response rates [45]. Personalising the email and appealing to a person's egotistic motivation are also shown to be strategies that improve online response rate [45,47]. For the current survey, recruitment may have been increased through direct emails to chairs or programme directors of research programmes rather than to generic email addresses or online forms. This may have also helped to ensure that the most appropriate people completed the survey, potentially increasing quality of responses and reducing need for exclusion.

The survey response rate corresponds to the ESF survey [31] which also received 30 responses from organisations in Europe and one in the USA. Our survey focus differed from the ESF survey [31], as we were mainly interested in the different types of peer review process that funders used, in order to better understand the current landscape of peer review, rather than questions on quality and management of peer review and peer reviewers more generally. Thus, our data contribute to a better understanding of decision-making processes and nuances of these processes that are being employed in current practice within health research funding organisations.

This study also provides limited quantitative data and there was the potential for researcher bias during data extraction. The researchers were mindful of their preconceptions about peer review practice (e.g. known challenges of bias and burden) and what peer review practices had been specifically asked about in the survey. Due to the nature of the survey questions, there was some grouping of codes under similar headings; however, this was not a formal predefined framework, the four main themes did not match questions, were discussed by the team and were triangulated with quantitative data. As such, we are confident that the results have been interpreted fairly.

In conclusion, given the emergence of innovative decision-making approaches, the aim of this study was to better understand current decision-making practices for the allocation of health research funding from the perspective of UK and international funders in order to determine what approaches were being utilised and why. The key findings from this survey show that similar decision-making processes tend to be used by all funders and there are many nuances and challenges to these processes. These processes are engaged in because it is considered the optimum way to make funding allocation decisions and demonstrates good practice. Funders continually strive for improvements in decision-making practice, and recognise the need to develop more flexible and supportive approaches that will facilitate decision-making (by reducing bias and burden) whilst maintaining key positive values such as transparency, fairness and quality. However, findings indicate a preference to adapt current systems rather than use innovative processes. This may be due to the lack of evidence available and/or the difficulties that trialling and testing new practices may cause. Thus, it is clear that more empirical studies are needed to evaluate the effectiveness of different peer review approaches, in order to provide funders with a sound evidence-base about what and how practices can be implemented to help inform decision-making in research fund allocation.

## Supporting information

**S1 File. The survey sent to funding organisations.** Some questions would only be shown depending on prior answers.
(DOCX)

**S1 Dataset. A subset of anonymised data received from the survey.** To maintain confidentiality the dataset is split into four sections (organisation demographics, assessment criteria and current peer review practice). All sections have been anonymised and randomised (so row 1 is not necessarily the same funder across all tabs).
(XLSX)

## Acknowledgments

We would like to thank Helen Payne for her advice on the project, members of the NIHR staff who piloted the survey, all of those who helped to disseminate the survey, all of the respondents of the survey and to those who commented on the final manuscript.

## Author Contributions

**Conceptualization:** Katie Meadmore, Kathryn Fackrell, Alejandra Recio-Saucedo, Abby Bull, Amanda Blatch-Jones.

**Data curation:** Katie Meadmore, Kathryn Fackrell, Alejandra Recio-Saucedo, Amanda Blatch-Jones.

**Formal analysis:** Katie Meadmore, Kathryn Fackrell, Alejandra Recio-Saucedo, Simon D. S. Fraser, Amanda Blatch-Jones.

**Investigation:** Katie Meadmore, Kathryn Fackrell, Alejandra Recio-Saucedo, Abby Bull, Amanda Blatch-Jones.

**Methodology:** Katie Meadmore, Kathryn Fackrell, Alejandra Recio-Saucedo, Abby Bull, Simon D. S. Fraser, Amanda Blatch-Jones.

**Project administration:** Katie Meadmore.

**Validation:** Katie Meadmore, Kathryn Fackrell, Alejandra Recio-Saucedo, Abby Bull, Simon D. S. Fraser, Amanda Blatch-Jones.

**Writing – original draft:** Katie Meadmore, Kathryn Fackrell.

**Writing – review & editing:** Katie Meadmore, Kathryn Fackrell, Alejandra Recio-Saucedo, Abby Bull, Simon D. S. Fraser, Amanda Blatch-Jones.

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
