## [Decision Letter · Decision Letter 0]

27 May 2020

PONE-D-20-03358

Decision-making approaches used by UK and international health funding organisations for allocating research funds: A survey of current practice

PLOS ONE

Dear Dr. Meadmore,

Thank you for submitting your manuscript to PLOS ONE. After careful consideration, we feel that it has merit but does not fully meet PLOS ONE’s publication criteria as it currently stands. Therefore, we invite you to submit a revised version of the manuscript that addresses the points raised during the review process.

We look forward to receiving your revised manuscript.

Kind regards,

Shelina Visram, PhD, MPH, BA

Academic Editor

PLOS ONE

Journal Requirements:

2. Please include your tables as part of your main manuscript and remove the individual files. Please note that supplementary tables should remain as separate "supporting information" files

Reviewers' comments:

Reviewer's Responses to Questions

**Comments to the Author**

1. Is the manuscript technically sound, and do the data support the conclusions?

Reviewer #1: Partly

Reviewer #2: Yes

2. Has the statistical analysis been performed appropriately and rigorously? 

Reviewer #1: I Don't Know

Reviewer #2: N/A

3. Have the authors made all data underlying the findings in their manuscript fully available?

Reviewer #1: No

Reviewer #2: No

4. Is the manuscript presented in an intelligible fashion and written in standard English?

Reviewer #1: Yes

Reviewer #2: Yes

5. Review Comments to the Author

Reviewer #1: Abstract – needs to refer more clearly to which types of funders were participants; for example, ‘other international countries’ is a little vague, can you be more explicit? The way in which the funders are referred to in the abstract suggests that they are relatively homogenous in terms of the types of health research that are funded – can you be more clear if the focus was, for example, biomedical health research, research on wider determinants of health? or both and more? I think that the abstract could also more clearly demonstrate the value and need for this research – which is this important to know? One of the conclusions is about a solid evidence base but could have been clearer how this came from the results. Also, I wasn’t clear what “reviewer diversity that were integral to current practice or important for future practice” meant – can you clarify?

Need for /value of this research needs to be more clearly demonstrated at the start of the piece. It is mentioned that difficult decisions need to be made – can you be more clear what these are? decisions between different types of health research? prevention vs treatment? biomedical vs wider determinants research? There are fundamental differences in ontology/epistemology within the field of health research that could for example at least be alluded to? Can you be more clear about some of the fundamental difficulties in the health field in particular? There is an assertion that there have been wider developments that have happened, mentioning “public contribution and new data legislations” – can you explain more clearly why these are significant, give some examples? And you indicate that peer review has undergone limited change but then go on to give some examples of change – this seems a little inconsistent. Can you be clearer what the need and value of this research is?

Methods – can you be more clear what you mean by wider reach – wider reach that what? (do you just mean a wide reach and where? in the UK, globally? Also, can you explain what you mean by phemenological approach? You refer to the survey questions being developed by a variety of stakeholders but these seem only to be NIHR / NIHR team members, which does not seem that various; I would recommend being more clear about this and how the survey was developed – how was previous research/literature used to develop the survey questions? What is a ‘think aloud pilot’? It would be useful earlier in the methods to indicate who the intended respondents of the survey were. Can you be more clear what you mean by data being screened for quality? What did this involve? Are you suggesting that some data was removed on this basis? How did wider literature on the topic inform the data analysis?

Results / discussion – some interesting findings are highlighted and points discussed. At times, there is use of terminology such as “right questions, teams and research proposals are funded” which suggests that there is a ‘right’ answer to what should be funded. It is not immediately clear that it is always the case that there is a ‘right’ answer – as decision-making involves making value-based choices between alternative, particularly in the health field where, for example, decisions might be made between biomedical research and health-related social science research – what would be the ‘right’ decision here? I think that some of the discussion needs to be more nuanced and reflective of this type of issue.

Earlier on the piece there is mention of a “broad range of health areas, including ageing, neurodegenerative diseases, cancer, diabetes, meningitis, health technology, HIV and AIDS, heart disease and stroke. They covered basic science through to applied clinical research as well as health service delivery, and included disease-specific programmes, public health and global health.” This breadth and complexity of the health field needs to be recognised more in the discussion I think; with some recognition that this relates also to your points about ‘expertise’ - if health funding covers all the above, it is of course challenging to cover all these areas? What other aspects of ‘expertise’ are also important? What about contextual knowledge – which can be particularly important, for example, in evaluative research/implementation research where context can shape how interventions are received/adopted/rolled out/scaled up? Recognition of some of these issues would show more reflection/consideration of how the data in the study is important?

Reviewer #2: Thank you for the opportunity to review this manuscript, which is well-written and has the potential to contribute to the literature on decision-making by research funders. I have a number of suggested revisions that I believe will improve the manuscript:

ABSTRACT: I find the use of the terms ‘current decision-making processes’ and ‘alternatives’ confusing here. If some are using these alternatives then surely they are also part of the current processes? The authors seem to be making some assumptions about the ‘normal’ way of doing things and more ‘unusual’ approaches. If peer review is seen as the ‘norm’ and other approaches are somehow more ‘innovative’ then this needs to be made clear in the abstract.

INTRODUCTION: The final sentence of the first paragraph (lines 57-60) needs to be supported by evidence/relevant citations.

METHODS: The survey development and piloting process appears to have been rigorous and is well-described. However, I would have liked to see the ‘think aloud’ pilot explained further, ideally with reference to the literature.

A minor point but the sentence at lines 144-146 (and the following sentence) could be re-worded to avoid using the words ‘organisations’ and ‘targeted’ multiple times.

76 health research organisations were identified – across how many countries? It is not entirely clear who potential respondents were, e.g. where emails were person-specific, were these targeted towards chairs of funding panels, administrative support staff, etc?

Lines 174-175: how exactly were multiple responses from the same organisation ‘merged’? Were any in conflict?

The description of the analytical process is confusing, particularly the suggestion that inconsistencies or limited responses were excluded. What exactly does this mean? It would be helpful if the authors could say more about this because it leaves the reader wondering if they excluded responses that did not fit with their a priori assumptions. How does this fit with the phenomenological approach (which is mentioned at lines 113-115) and thematic analysis?

Figure 1 – providing a textbook illustration of the thematic analysis process – is not necessary.

RESULTS: Related to the point above about excluding certain responses, I am not clear why only 31 of 35 responses were included (resulting in the exclusion of 1 of 24 organisations). Figure 2 is helpful but does not make clear what exclusion on the basis of quality means. It is however very interesting that the non-targeted recruitment resulted in few additional responses; this is an important learning point for other researchers that I think the authors should pick up on in their discussion.

Much of the information in the sub-section on respondent characteristics might be better presented in the form of a table.

Using n for number of responses is confusing where this actually relates to the number of organisations, e.g. lines 221-223. This also applies when reporting qualitative findings; I am really not clear what n=28 means at line 230. Presumably this denotes the number of usable survey responses that included qualitative data, but this is not made clear in the text.

I was surprised not to see quotes attributed by country or any discussion of inter-country differences in the survey findings.

Redaction of the quote at lines 372-373 has obscured the meaning. Please add some descriptive text to indicate what [name] relates to, i.e. is it an organisation, a process, etc?

DISCUSSION: The discussion is good, with clear recommendations. As stated above, I would like to see some discussion on any inter-country differences and also lessons learned for researchers conducting similar studies.

6. PLOS authors have the option to publish the peer review history of their article (what does this mean?). If published, this will include your full peer review and any attached files.

Reviewer #1: No

Reviewer #2: No

---

## [Author Response · Author response to Decision Letter 0]

3 Jul 2020

Comments to the Author

1. Is the manuscript technically sound, and do the data support the conclusions?

Reviewer #1: Partly

Reviewer #2: Yes

Response: We have addressed reviewer 1’s comments and hope that the revisions have provided further clarity to the conclusions drawn________________________________________

2. Has the statistical analysis been performed appropriately and rigorously? 

Reviewer #1: I Don't Know

Reviewer #2: N/A

We have addressed the reviewers’ comments and hope that the revisions have provided further clarity to the methods used throughout the study.

3. Have the authors made all data underlying the findings in their manuscript fully available?

Reviewer #1: No

Reviewer #2: No

Response: We have included all quantitative data to supplement the results reported (see S2 datasets). This has been split into different sections in order to ensure that we maintain confidentiality and adhere to our ethical protocol.________________________________________

 4. Is the manuscript presented in an intelligible fashion and written in standard English?

Reviewer #1: Yes

Reviewer #2: Yes

Response: We are very pleased that the reviewers found the manuscript to be written in an intelligible way. 

 5. Review Comments to the Author

Reviewer #1: 

1.1. Abstract – needs to refer more clearly to which types of funders were participants; for example, ‘other international countries’ is a little vague, can you be more explicit? 

Response: This was originally written so as not to specifically name individual countries to maintain confidentiality. However, on reflection, we agree with the reviewer and the current wording is vague and potentially misleading. We feel that we can be more explicit here without breaking confidentiality, as this information is not linked to any other identifying information. This has now been changed.

1.2. The way in which the funders are referred to in the abstract suggests that they are relatively homogenous in terms of the types of health research that are funded – can you be more clear if the focus was, for example, biomedical health research, research on wider determinants of health? or both and more? 

Response: The focus was left broad as we wanted any funder related to the health sector to be able to respond and so the remit encompassed biomedical and wider determinants of health too. We have made this clearer in the abstract: “An online survey (active March-April 2019) was distributed by email to UK and international health and health-related funding organisations (e.g., biomedical and social)”

1.3. I think that the abstract could also more clearly demonstrate the value and need for this research – which is this important to know? 

Response: Thank you for this comment, a sentence has been added to the objectives: “Innovations in decision-making practice for allocation of funds in health research are emerging; however, it is not clear to what extent these are used.”

1.4. One of the conclusions is about a solid evidence base but could have been clearer how this came from the results. 

Response: The conclusion was drawn from the findings that funders are not averse to innovative or different approaches to decision making but that they wanted more information on these. We inferred from this that an evidence base was required. We have now revised this sentence and removed this phrase: “Funders indicated that they wanted more information and empirical evidence on innovative approaches which would help to inform decision-making in research fund allocation”

1.5. Also, I wasn’t clear what “reviewer diversity that were integral to current practice or important for future practice” meant – can you clarify?

Response: This sentence referred to the values that underlie why funders engage in particular practices. We agree that reviewer diversity in the current sentence was confusing and so this has been re-worded: “Key values underpinned decision-making processes. These included transparency and gaining perspectives from reviewers with different expertise (e.g., scientific, patient and public)”

1.6. Need for /value of this research needs to be more clearly demonstrated at the start of the piece. 

Response: We have added some additional sentences to demonstrate need of this research to the first paragraph of the introduction: “The challenges to funding decision-making are well established and numerous, and although innovations in decision-making are emerging, it is not clear to what extent these are utilised by funders.”

1.7. It is mentioned that difficult decisions need to be made – can you be more clear what these are? decisions between different types of health research? prevention vs treatment? biomedical vs wider determinants research? There are fundamental differences in ontology/epistemology within the field of health research that could for example at least be alluded to? Can you be more clear about some of the fundamental difficulties in the health field in particular? 

Response: We have added some examples specific to health research decisions to the first paragraph of the introduction: “For example, deciding which health areas have priority, and whether research may lead to changes in practice and better health outcomes (be that patient, economic or social benefit).”

1.8. There is an assertion that there have been wider developments that have happened, mentioning “public contribution and new data legislations” – can you explain more clearly why these are significant, give some examples? And you indicate that peer review has undergone limited change but then go on to give some examples of change – this seems a little inconsistent. Can you be clearer what the need and value of this research is?

Response: We have revised the final two paragraphs of the introduction to provide a clearer description of the value of the research. We know that there are challenges to decision-making, including bias towards innovation or early career researchers, and although innovations in decision-making are emerging, these are limited in terms of empirical evidence. In addition, it is not clear to what extent these are utilised by funders. Identifying what approaches are utilised in current practice in decision-making and whether they include innovative approaches, may provide better understanding about decision-making and why funders are engaged (or not) in exploring mechanisms to change/enhance funding processes to address known challenges.

1.9. Methods – can you be more clear what you mean by wider reach – wider reach that what? (do you just mean a wide reach and where? in the UK, globally? 

Response: By wider reach, we meant that an online survey compared to a paper survey might be more likely to be picked up globally and by organisations that we did not individually target. We have amended this to read national and global reach.

1.10. Also, can you explain what you mean by phemenological approach? 

Response: A phenomenological approach is an approach used in qualitative research that focuses on describing something by exploring it from the perspective of those that have experienced it – in this case, describing decision processes from the perspectives of the research programmes in organisations funding health and health related research.

1.11. You refer to the survey questions being developed by a variety of stakeholders but these seem only to be NIHR / NIHR team members, which does not seem that various; I would recommend being more clear about this and how the survey was developed – how was previous research/literature used to develop the survey questions? 

Response: We have removed “variety of stakeholders” so that there is no ambiguity about developing the survey with anybody external to NIHR. We have also revised the text so that it is clear how we used previous research to develop the survey questions.

1.12. What is a ‘think aloud pilot’?

Response: By think aloud pilot, we were referring to a form of verbal feedback. A researcher sat with the staff member while they verbally described their thoughts about the questionnaire (the questions, answer choices, layout, wording etc) as they went through it. The researcher noted thoughts, comments and questions and engaged in dialogue with the member of staff to determine validity and ease of use of the questionnaire. To reduce confusion, we have removed this term and changed it to verbal feedback.

1.13. It would be useful earlier in the methods to indicate who the intended respondents of the survey were. 

Response: The intended respondents have been included in the first paragraph of the methods: “This study used a survey design to gather quantitative and qualitative information about decision-making practices used by organisations that fund health and health-related research”

1.14. Can you be more clear what you mean by data being screened for quality? What did this involve? Are you suggesting that some data was removed on this basis? How did wider literature on the topic inform the data analysis?

Response: For two of the responses, many answers included “don’t know” or “not sure” or were not answered. In addition, for these two responses, the respondents also left comments to say that they did not think that they were the best person to complete the survey. As such, it was decided that the survey responses for these two respondents would not be analysed as they may not provide a true reflection of the organisation processes. This has been made clearer in the results.

1.15. Results / discussion – some interesting findings are highlighted and points discussed. 

Response: Thank you for this comment.

1.16. At times, there is use of terminology such as “right questions, teams and research proposals are funded” which suggests that there is a ‘right’ answer to what should be funded. It is not immediately clear that it is always the case that there is a ‘right’ answer – as decision-making involves making value-based choices between alternative, particularly in the health field where, for example, decisions might be made between biomedical research and health-related social science research – what would be the ‘right’ decision here? I think that some of the discussion needs to be more nuanced and reflective of this type of issue.

Response: Thank you for bringing this to our attention. We completely agree that there is no right decision and had not intended to convey this. We have amended this phrase in the results and discussion, and have included some additional text in the discussion to reflect this.

1.17. Earlier on the piece there is mention of a “broad range of health areas, including ageing, neurodegenerative diseases, cancer, diabetes, meningitis, health technology, HIV and AIDS, heart disease and stroke. They covered basic science through to applied clinical research as well as health service delivery, and included disease-specific programmes, public health and global health.” This breadth and complexity of the health field needs to be recognised more in the discussion I think; with some recognition that this relates also to your points about ‘expertise’ - if health funding covers all the above, it is of course challenging to cover all these areas? What other aspects of ‘expertise’ are also important? What about contextual knowledge – which can be particularly important, for example, in evaluative research/implementation research where context can shape how interventions are received/adopted/rolled out/scaled up? Recognition of some of these issues would show more reflection/consideration of how the data in the study is important?

Response: Thank you for this comment. We have included some additional text in the discussion to show how funding organisations have to balance getting the appropriate expert reviewers that align with their strategic aims and goals and the scientific field of the application whilst also trying to contain burden and bias. As you suggest, different funding organisations have different priorities and resource which influences how this is done: “However, these solutions also come with their own challenges. For example, health research covers a broad range of topics and so what is considered expertise will also differ across applications and funders. In practice, funding organisations cannot feasibly cover all areas for each application (especially when trying to reduce burden) and therefore need to balance number of reviewers with which areas of expertise are priority.”

2.1 Reviewer #2: Thank you for the opportunity to review this manuscript, which is well-written and has the potential to contribute to the literature on decision-making by research funders. I have a number of suggested revisions that I believe will improve the manuscript:

Response: Thank you for your comments, they have been very helpful.

2.2 ABSTRACT: I find the use of the terms ‘current decision-making processes’ and ‘alternatives’ confusing here. If some are using these alternatives then surely they are also part of the current processes? The authors seem to be making some assumptions about the ‘normal’ way of doing things and more ‘unusual’ approaches. If peer review is seen as the ‘norm’ and other approaches are somehow more ‘innovative’ then this needs to be made clear in the abstract.

Response: Thank you for this comment. This was not our intention but we can see how the terminology used here may be interpreted in this way. We have revised the abstract and manuscript more generally to change ‘alternatives’ to innovative approaches, where appropriate and made it clear that triage, external peer review and committee meetings were the typical decision-making practice.

2.3. INTRODUCTION: The final sentence of the first paragraph (lines 57-60) needs to be supported by evidence/relevant citations. 

Response: A reference has been added.

2.4 METHODS: The survey development and piloting process appears to have been rigorous and is well-described. However, I would have liked to see the ‘think aloud’ pilot explained further, ideally with reference to the literature.

Response: In line with comment 1.12, we have now removed this phrase from the manuscript.

2.5. A minor point but the sentence at lines 144-146 (and the following sentence) could be re-worded to avoid using the words ‘organisations’ and ‘targeted’ multiple times.

Response: We had not spotted this before and agree that it should be re-worded. This has been done.

2.6. 76 health research organisations were identified – across how many countries?

Response: We have included the number of countries in the text: “Purposive and snowball sampling was used to recruit respondents. In order to obtain a broad funder perspective 76 health research organisations (109 emails) were contacted across 10 different countries”.

2.7. It is not entirely clear who potential respondents were, e.g. where emails were person-specific, were these targeted towards chairs of funding panels, administrative support staff, etc?

Response: More detail has been added about the targeted emails: “Where possible emails were sent named administrative staff for specific research programmes, but also included general enquiry email addresses and online forms.”

2.8. Lines 174-175: how exactly were multiple responses from the same organisation ‘merged’? Were any in conflict?

Response: Multiple responses were merged so that there was only one entry for each research programme. The most complete entry was kept, and any blank responses were filled using the other entry. Open responses were added together, and close question conflicts (which were minimal; e.g., number of staff) were resolved through discussion and by checking the funder website. This has been added to the text.

2.9. The description of the analytical process is confusing, particularly the suggestion that inconsistencies or limited responses were excluded. What exactly does this mean? It would be helpful if the authors could say more about this because it leaves the reader wondering if they excluded responses that did not fit with their a priori assumptions. 

Response: Thank you for bringing this to our attention. Consistency was the wrong word to use here. Data was only merged for multiple responses from the same research programme so there was only one data entry per research programme. For two of the responses, many answers included “don’t know” or “not sure” or were not answered. In addition, for these two responses, the respondents left comments to say that they did not think that they were the best person to complete the survey. As such, it was decided that the survey responses for these two respondents would not be analysed as they may not provide a true reflection of the organisation processes. Three further responses were not used for the qualitative analysis as no open questions had been answered. Qualitative data analysis was inductive and we had no a priori assumptions. We have provided more context around data exclusion. 

2.10. How does this fit with the phenomenological approach (which is mentioned at lines 113-115) and thematic analysis? 

Response: Data preparation occurred before the analysis was undertaken and only those respondents who did not answer any open questions were excluded from the qualitative analysis as there was no data to code.

2.11. Figure 1 – providing a textbook illustration of the thematic analysis process – is not necessary.

Response: We have removed this figure from the manuscript.

2.12. RESULTS: Related to the point above about excluding certain responses, I am not clear why only 31 of 35 responses were included (resulting in the exclusion of 1 of 24 organisations). 

Response: More detail about the exclusion process has been added to the manuscript and figure caption.

2.13. Figure 2 is helpful but does not make clear what exclusion on the basis of quality means. 

Response: More detail about the exclusion process has been added to the manuscript and figure caption.

2.14. It is however very interesting that the non-targeted recruitment resulted in few additional responses; this is an important learning point for other researchers that I think the authors should pick up on in their discussion.

Response: Thank you for this comment. We have included this as a learning point in the discussion: “Note also that five respondents were recruited via the use of e-promotion and promotion at conferences, demonstrating the benefit of multi-modal recruitment routes (46). Multi-modal recruitment methods are recommended to researchers looking to recruit international or multidisciplinary samples (46). We also sent reminders and stated the average time it would take to complete the survey. Both strategies are suggested to improve online survey response rates (45). Personalising the email and appealing to a person’s egotistic motivation are also shown to be strategies that improve online response rate (45,47). For the current survey, recruitment may have been increased through direct emails to chairs or programme directors of research programmes rather than to generic email addresses or online forms. This may have also helped to ensure that the most appropriate people completed the survey, potentially increasing quality of responses and reducing need for exclusion.”

2.15. Much of the information in the sub-section on respondent characteristics might be better presented in the form of a table.

Response: We did initially have this information in a table but found that written text worked better, as such we have not reverted back to a table format.

2.16. Using n for number of responses is confusing where this actually relates to the number of organisations, e.g. lines 221-223. This also applies when reporting qualitative findings; I am really not clear what n=28 means at line 230. Presumably this denotes the number of usable survey responses that included qualitative data, but this is not made clear in the text.

Response: We have removed n from this section and replaced with the actual numbers. We have also removed n=28 and explained earlier on the text that qualitative analysis was conducted on 28 responses. 

2.17. I was surprised not to see quotes attributed by country or any discussion of inter-country differences in the survey findings.

Response: We agree that inter-country differences would be really interesting to explore. However, the intention of the study was never to explore inter-country differences but instead just to describe what decision-making practices were being used more generally across countries. In addition, given the small numbers of responses we did not think that comparisons were appropriate. In order to remain true to our original research questions, we have not conducted any further analysis on inter-country difference. 

2.18. Redaction of the quote at lines 372-373 has obscured the meaning. Please add some descriptive text to indicate what [name] relates to, i.e. is it an organisation, a process, etc?

Response: Thank you for pointing this out. [name] referred to an organisation and so we have added this context to the quote.

2.19. DISCUSSION: The discussion is good, with clear recommendations. As stated above, I would like to see some discussion on any inter-country differences and also lessons learned for researchers conducting similar studies.

Response: As described in 2.17, as we did not set out to make comparisons across countries and due to the small number of responses we have decided not to conduct any further inter-country analysis. However, in line with comment 2.14, we have included some discussion on lessons learned for recruitment.

---

## [Decision Letter · Decision Letter 1]

19 Aug 2020

PONE-D-20-03358R1

Decision-making approaches used by UK and international health funding organisations for allocating research funds: A survey of current practice

PLOS ONE

Dear Dr. Meadmore,

Thank you for submitting your manuscript to PLOS ONE. After careful consideration, we feel that it has merit but does not fully meet PLOS ONE’s publication criteria as it currently stands. Therefore, we invite you to submit a revised version of the manuscript that addresses the points raised during the review process.

ACADEMIC EDITOR: Thank you for making the revisions to this paper. R1 is happy that all previous comments have been addressed. Unfortunately the original R2 was not available and so a new reviewer was approached to look at the revised manuscript. They have suggested an additional minor change/clarification that should be very quick and easy to address, and should help to make description of the study design more rigorous. 

We look forward to receiving your revised manuscript.

Kind regards,

Shelina Visram, PhD, MPH, BA

Academic Editor

PLOS ONE

Reviewers' comments:

Reviewer's Responses to Questions

**Comments to the Author**

1. If the authors have adequately addressed your comments raised in a previous round of review and you feel that this manuscript is now acceptable for publication, you may indicate that here to bypass the “Comments to the Author” section, enter your conflict of interest statement in the “Confidential to Editor” section, and submit your "Accept" recommendation.

Reviewer #2: All comments have been addressed

Reviewer #3: All comments have been addressed

2. Is the manuscript technically sound, and do the data support the conclusions?

Reviewer #2: (No Response)

Reviewer #3: Yes

3. Has the statistical analysis been performed appropriately and rigorously? 

Reviewer #2: (No Response)

Reviewer #3: Yes

4. Have the authors made all data underlying the findings in their manuscript fully available?

Reviewer #2: (No Response)

Reviewer #3: Yes

5. Is the manuscript presented in an intelligible fashion and written in standard English?

Reviewer #2: (No Response)

Reviewer #3: Yes

6. Review Comments to the Author

Reviewer #2: (No Response)

Reviewer #3: Thank you for the opportunity to review this paper. It is well written, methodologically sound and discusses an engaging issue well.

The comments to the author have been clearly addressed and result in a more rigorous paper.

Apologies for adding additional feedback, but one minor issue could be discussed in order to ensure full rigour. Your paper states that you took a mixed methods approach to data analysis, but then give only detail of the analysis for qualitative and quantitative strands as separate entities. There is no discussion of synthesis of the strands. If this was carried out, can it be detailed (e.g. was the analysis sequential or prioritised in any way)? If it was not, can this absence be justified in relation to your research question and design?

Thanks again - and apologies for the minor revisions.

7. PLOS authors have the option to publish the peer review history of their article (what does this mean?). If published, this will include your full peer review and any attached files.

Reviewer #2: No

Reviewer #3: No

---

## [Author Response · Author response to Decision Letter 1]

7 Sep 2020

6. Review Comments to the Author

Reviewer #3: Thank you for the opportunity to review this paper. It is well written, methodologically sound and discusses an engaging issue well.

The comments to the author have been clearly addressed and result in a more rigorous paper.

Apologies for adding additional feedback, but one minor issue could be discussed in order to ensure full rigour. Your paper states that you took a mixed methods approach to data analysis, but then give only detail of the analysis for qualitative and quantitative strands as separate entities. There is no discussion of synthesis of the strands. If this was carried out, can it be detailed (e.g. was the analysis sequential or prioritised in any way)? If it was not, can this absence be justified in relation to your research question and design?

Response: Thank you for taking the time to review our manuscript and for your comment. We agree that this detail was missing and have added in some text to clarify how we approached the mixed methods analysis in the data analysis section of the methods: “A mixed methods approach to the analysis was taken in order to cross-validate findings and provide a fuller picture of the peer review landscape. Data were analysed separately but concurrently, and findings from each strand of data were triangulated to inform and explain patterns and interpretation. For example, qualitative data was used to expand and interpret quantitative findings and frequency data was used to confirm patterns in qualitative data.”

---

## [Editor Report · Decision Letter 2]

14 Sep 2020

Decision-making approaches used by UK and international health funding organisations for allocating research funds: A survey of current practice

PONE-D-20-03358R2

Dear Dr. Meadmore,

We’re pleased to inform you that your manuscript has been judged scientifically suitable for publication and will be formally accepted for publication once it meets all outstanding technical requirements.

Kind regards,

Shelina Visram, PhD, MPH, BA

Academic Editor

PLOS ONE
---

## [Editor Report · Acceptance letter]

26 Oct 2020

PONE-D-20-03358R2 

Decision-making approaches used by UK and international health funding organisations for allocating research funds: A survey of current practice 

Dear Dr. Meadmore:

I'm pleased to inform you that your manuscript has been deemed suitable for publication in PLOS ONE. Congratulations! Your manuscript is now with our production department. 

Kind regards, 

on behalf of

Dr. Shelina Visram 

Academic Editor

PLOS ONE